# Clinical Performance of Cobas 6800 for the Detection of High-Risk Human Papillomavirus in Urine Samples

**DOI:** 10.3390/vaccines11061071

**Published:** 2023-06-06

**Authors:** Brian Joseph Hajjar, Ummar Raheel, Rachel Manina, Jovanie Simpson, Muhammad Irfan, Yasir Waheed

**Affiliations:** 1Telostrand Innovations LLC, Hackensack, NJ 07601, USA; 2Office of Research, Innovation & Commercialization, Shaheed Zulfiqar Ali Bhutto Medical University (SZABMU), Islamabad 44000, Pakistan; 3Gilbert and Rose-Marie Chagoury School of Medicine, Lebanese American University, Byblos 1401, Lebanon

**Keywords:** HPV, cervical cancer, Cobas 6800, urine testing, gender-neutral screening

## Abstract

Testing for high-risk human papillomavirus (HPV) as part of primary cervical cancer screening has become more common recently. The Cobas 6800, an FDA-approved cervical screening platform, detects 14 high-risk HPVs, including HPV16 and HPV18. However, this test is limited to only women, which leads to low screening rates in trans men and other non-binary people. The cervical screening of trans men and other genders, especially those lying on the female-to-male spectrum, is equally important. Furthermore, cisgender males, particularly homosexuals, are also prone to chronic HPV infections and serve as HPV carriers, transmitting it to women and other men through sexual contact. Another limitation of the test is its invasive specimen collection, which induces discomfort and genital dysphoria. Therefore, there is a need for an innovative, less invasive method that would allow the sampling process to be more comfortable. In this study, we assess the performance of the Cobas 6800 for high-risk HPV detection in urine samples spiked with HPV16, HPV18, and HPV68. The limit of detection (LOD) was calculated using a dilution series (1.25–10,000 copies/mL) over a course of three days. Furthermore, the clinical validation was performed by calculating sensitivity, specificity, and accuracy. The limit of detection ranged from 50–1000 copies/mL depending upon the genotype. Moreover, the urine test demonstrated a high clinical sensitivity of 93%, 94%, and 90% for HPV16, HPV18, and HPV68, with 100% specificity. The overall percent agreement was calculated to be 95% for both HPV16 and HPV18, and 93% for HPV68. The high concordance, reproducibility, and clinical performance of the current assay suggest that the urine-based HPV test fulfills the requirements for its use in primary cervical screening. Moreover, it has the potential to be used for mass screening to not only identify high-risk individuals, but also to monitor vaccine effectiveness.

## 1. Introduction

Human Papillomavirus (HPV), a member of *Papillomaviridae,* is a non-enveloped double-stranded DNA virus which is classified into low-risk HPVs involved in causing anogenital and cutaneous warts, and high-risk HPVs (16, 18, 31, 33, 35, 39, 45, 51, 52, 56, 58, 59, 66, 68) that are oncogenic and whose persistent infections are responsible for oropharyngeal and anogenital cancers [1,2,3]. Most HPV infections are asymptomatic, and about 90% of these are cleared by the immune system itself [4,5]. The remaining 10–12% of the infections that are caused by high-risk HPV lead to invasive cancers [6,7].

Cervical cancer is the most common type of HPV-induced cancer worldwide [1], and is the third most common cancer [8], responsible for the highest mortality in women, after breast cancer [9]. About 604,000 HPV-related cases of cervical cancer were reported in 2020 alone, and HPV was found to be responsible for 341,000 cervical-cancer-related deaths [10]. Since most HPV infections do not show any symptoms, routine screening is essential for the prevention and treatment of sexually transmitted infections (STIs). The number of cases of cervical cancer may be reduced by organized screening programs. Moreover, HPV-based screening is more effective than cytology screening in preventing cervical cancer in large randomized controlled studies [11,12].

The Cobas 6800 is an automated high-throughput FDA-approved platform used for routine cervical cancer screening. It uses a quantitative polymerase chain reaction (PCR) to detect 14 types of high-risk HPV along with HPV16 and HPV18. The assay starts with automated nucleic acid extraction from the clinician-obtained samples followed by its binding with magnetic glass particles. The purified nucleic acid is then amplified using a thermostable DNA polymerase. The amplification of sequences of high-risk HPVs and internal control β-globin occurs simultaneously, using specific primers and probes. The automated management software assigns the results for the tests as positive, negative, or invalid [13,14]. The Cobas 6800 is a sensitive HPV detection system, but its utility is limited by the invasive nature of the sampling procedures and the fact that it targets only women. It has been estimated that about 73% of the women attend the screening programs and half of cervical cancer cases are reported among the women who are not willing to attend due to discomfort and embarrassment [15]. Moreover, the female-focused screening misses the fact that both cis and trans men act as carriers for the transmission of HPV to women and other men through sexual intercourse [16]. Furthermore, the rates of cervical cancer screening in transgender men and other non-binary people with an intact cervix is very low compared with cisgender women despite comparable susceptibility to HPV infection [17,18,19,20]. The main reason behind this low rate of screening is the discomfort and genital dysphoria associated with anxiety-provoking pelvic examinations [21,22]. Keeping in view these challenges and limitations, it is necessary to develop and use non-invasive, less painful, and gender-neutral screening approaches, such as urine testing, for detecting HPV infections to circumvent these barriers [23]. The detection of HPV in urine samples is a non-invasive and cost-effective method, which is well accepted by women and trans people in the United States [24,25,26]. The first-void urine testing also has the potential to increase the rates of HPV screening, especially in non-binary people who prefer self-sampling and are not willing to go through an invasive speculum examination [27,28]. In this study, we aim to develop a Cobas 6800-based non-invasive and gender-inclusive test for HPV detection in first-void urine samples.

## 2. Materials and Methods

### 2.1. Study Design

The study was performed on a total of 40 urine samples, out of which 10 were negative controls and 30 were spiked positive samples. All the participants, after filling out the informed consent, provided self-collected first-void urine samples which were stored at 2–27 °C. The study was aimed at finding the LOD and evaluating the sensitivity, specificity, reproducibility, and clinical accuracy of the Cobas 6800 for HPV detection using urine samples. All the experiments were carried out at Telostrand Laboratory, LLC, in Hackensack, NJ, USA, a Clinical Laboratory Improvement Amendments (CLIA)-accredited clinical laboratory for high-complexity testing.

### 2.2. Clinical Samples and HPV Detection Using Cobas 6800

The study included 30 positive urine samples, which were spiked with HPV 16, 18, and 68 virus, and 10 negative samples with no HPV. The positive samples were spiked at different concentrations of the HPV whole organism controls (Exact Diagnostics LLC, Fort Worth, TX, USA) based on LOD data ranging from 1.25 copies/mL to 10,000 copies/mL with a final volume of 1 mL. All the positive spiked and negative urine samples were run on the Cobas 6800 according to the manufacturer’s instructions in triplicate over a period of three days to examine the sensitivity, specificity, percent agreement, and accuracy of the assay. The Cobas 6800 is an automated high-throughput real-time PCR-based system, which is used for routine cervical cancer screening and detects 14 types of high-risk HPV (HPV-31, -33, -35, -39, -45, -51, -52, -56, -58, -59, -66, and -68) along with HPV16 and HPV18. The assay starts with the extraction and release of nucleic acid from the samples, which then binds to the silica surface of the added magnetic glass particles. The purified nucleic acid is then amplified by a PCR reaction using thermostable DNA polymerase. The HPV and β-globin sequences are amplified simultaneously using a universal PCR amplification profile with predefined temperature steps and number of cycles. The automated management software assigns the results for the tests as positive, negative, or invalid [14,15]. In order to assess the sensitivity of the HPV detection test in real urine samples from individuals with a known history of urinary and reproductive health concerns, a pilot study was carried out. The study involved 137 male and 63 female urine samples, and the HPV detection was carried out using similar conditions.

### 2.3. LOD Studies

The lowest concentration detected in the three replicates was determined as the LOD. The LOD was established with dilution series of 10,000 copies/mL, 5000 copies/mL, 1000 copies/mL, 500 copies/mL, 250 copies/mL, 125 copies/mL, 50 copies/mL, 25 copies/mL, 5 copies/mL, 2.5 copies/mL, and 1.25 copies/mL using whole organism HPV 16, 18, and 68 (Exact Diagnostics LLC) along with internal positive and negative, non-template, and negative extraction controls. All the samples were run thrice, on three different days. The Ct values for assay replicates were averaged, and the values were plotted. The LOD was determined to be between 1.25 and 10,000 copies/mL, depending on the uniformity of the Ct value of the detectable copy number.

### 2.4. Stability and Reproducibility Analysis

The stability and reproducibility of the assay using urine samples was analyzed with a total of 20 samples, including 5 HPV-negative and 15 HPV-spiked positive samples over a course of seven days. The stability was measured based on the detection of β globulin, and the urine sample was considered valid as long as the β globulin was detectable according to the protocol provided by the manufacturer, Roche Diagnostics. The reproducibility of the assay was assessed by calculating the coefficient of variation (CV%).

### 2.5. Comparative Analysis of ThinPrep and Urine-Based HPV Detection

The diagnostic accuracy of the Cobas 6800 system for identifying HPV types 16, 18, and 68 in urine was compared with that for identifying ThinPrep samples. First-void urine and ThinPrep samples were obtained from participants for analysis. For both sample types, the sensitivity, specificity, accuracy, and percentage of agreement were determined.

### 2.6. Statistical Analysis

The clinical sensitivity, specificity, overall percent agreement and accuracy were evaluated for all the samples and calculated as percentages. The Ct values were expressed as mean and standard deviation.

## 3. Results

### 3.1. LOD Analysis

Valid test results were obtained for all the positive spiked and negative urine samples using the Cobas 6800. The LOD for HPV16 and HPV18 was established at 50 copies/mL with three out of three positive calls with an average Ct value of 37.03 ± 1.37 and 35.56 ± 0.8, respectively. Similarly, the LOD for HPV68 was measured to be 1000 copies/mL with three out of three positive calls with an average Ct cut-off value of 33.03 ± 0. 31.

No HPV was detected for the negative controls, and the overall concordance was found to be 100% for all the urine samples tested. The LOD results of HPV16, HPV18, and HPV68 plotted against their respective mean Ct values are given in Figure 1.

### 3.2. Analysis of Clinical Performance

The clinical performance of the Cobas 6800 on urine samples was evaluated by calculating the sensitivity, specificity, accuracy, and overall percent agreement. The validation study was performed over three days for each positive and negative sample to evaluate the clinical performance of the assay. For HPV16, the sensitivity was calculated to be 93% with 100% specificity, as no false positive was recorded. Similarly, the sensitivity for HPV18 was established to be 94% with 100% specificity. For HPV68, the sensitivity was calculated to be 90% and the specificity was 100%, similar to that of HPV16 and HPV18. The overall percent agreement and accuracy for HPV16 and HPV18 were calculated to be 95%. However, both the overall percent agreement and accuracy were 93% for HPV68. The results of the clinical performance of the assay are given in Table 1.

A pilot study on urine samples of patients with a known history of reproductive and urinary health issues was also carried out to assess the sensitivity of the assay for HPV detection in real urine samples. The study included 137 male and 63 female urine samples. The HPV detection was performed on clinical urine samples using similar conditions. The assay showed positive results for HPV in 2 out of 137 male patients and 7 out of 63 female patients. The results of HPV detection in patients’ urine samples are given in Table 2.

### 3.3. Stability and Reproducibility

The stability study was performed over a course of 7 days on 20 urine samples, of which 5 were negative, and 15 were positive samples spiked with HPV16, HPV18, and HPV68. The samples were stored at 2–8 °C. All 20 samples were run on the Cobas 6800, and the Ct values were recorded and plotted for the complete duration of the stability experiment. For HPV16, there was a shift of approximately four Ct values observed, whereas for HPV18 and HPV68, a shift of two Ct values was observed. The coefficient of variation (CV%) was calculated as between 4.2 and 5.1% for HPV16, 1.9 and 3.6% for HPV18, 2.3 and 3.6% for HPV68, and 1.2 and 4.4% for β globulin, which lies within the acceptable range. For β globulin, this shift was four Ct values. The day 5 stability data for samples 1–5 was lost due to instrument malfunction. The results of stability analyses are given in Table 3.

### 3.4. Comparison of ThinPrep and Urine-Based HPV Detection

In one of the experimental studies, we compared the diagnostic parameters for detecting HPV16, 18, and 68 in urine and ThinPrep samples (classical/traditional method). It was found that the sensitivity of urine-based HPV detection using the Cobas 6800 for these HPV types ranged from 90% to 94%, whereas the sensitivity for ThinPrep was 100% for all three types. The calculated specificity for both sample types was found to be 100%. Additionally, we evaluated the accuracy and overall percent agreement for urine, which ranged from 93% to 95%, in comparison with 100% for ThinPrep. The results of comparison study are represented in Table 4. Based on the results, we can conclude that both methods, urine-based HPV detection using the Cobas 6800 and the ThinPrep collection method, exhibited comparable sensitivity, specificity, accuracy, and overall percent agreement.

## 4. Discussion

Screening for cervical cancer has entered a new era, wherein cytology-based examinations are replaced by more sensitive and robust hrHPV DNA testing, whereas cytology is now only performed for triage purposes among women who test positive for HPV. The Cobas 6800 is an automated PCR-based sensitive HPV detection platform using clinician-obtained cervical samples. The test uses DNA amplification and hybridization to detect 14 high-risk HPV genotypes, including HPV16 and HPV18 [29,30]. The sampling process for HPV detection using the Cobas 6800 is often found to be uncomfortable, and women, especially young girls, feel reluctant and embarrassed to go through Pap smears and vaginal examinations [15]. Another major limitation of Cobas HPV testing is that it is gender-specific and mainly targets women. However, HPV screening is equally important for males and non-binary people, especially those with an intact cervix. A screening method that relies on non-invasive self-sampling techniques would not only eliminate the need for a pelvic examination, but it would also eliminate the requirement of a health infrastructure to collect relevant biological samples. The first-void urine-based testing for high-risk HPV detection provides a convenient and less-invasive alternative to the Pap test [27,31]. Several studies have investigated urine-based high-risk HPV detection in an attempt to provide a non-invasive and well-accepted alternative for trans men and other non-binary people who are reluctant to take part in screening programs due to embarrassment and genital dysphoria [24,28,32,33,34]. These factors further support the use of HPV detection in urine as a cervical cancer screening method. In the present study, we optimized and assessed the performance of the Cobas 6800 automated screening platform for HPV detection using spiked urine samples. The Cobas 6800 is an FDA-approved PCR-based HPV detection platform only for clinician-obtained cervical samples, and no study has been performed yet to validate this test for urine samples.

The results of clinical performance showed that the Cobas 6800 performed very well and is compatible with urine samples for HPV detection. The LOD study was performed using urine samples spiked with HPV at a concentration ranging from 10,000–1.25 copies/mL. The findings revealed that the Cobas 6800 can effectively detect both HPV16 and HPV18 at a minimum concentration of 50 copies/mL with an average Ct value of 37 and 35.6, respectively, demonstrating higher sensitivity than the previously recorded results [35]. Moreover, both of the HPV genotypes produced two out of three positive calls at a concentration of 25 copies/mL, which were considered discordant results. These genotypes were completely undetected at much lower copy numbers such as 5, 2.5, and 1.25 copies/mL. The LOD for HPV68, however, was established at 1000 copies/mL with an average Ct value of 33. There were two out of three and one out of three positive calls recorded at lower copy numbers, i.e., 500 and 250 copies/mL, respectively. Previous studies have also shown comparatively less sensitivity and insufficient amplification of HPV68, which can be attributed to the low prevalence and sequence variance in the open reading frame (ORF) of the E6, E7, and L1 genes of HPV68 [36,37]. The internal positive control was also run with all three genotypes and showed high concordance, with an average Ct value of 35. Moreover, no HPV was detected for any of the negative controls. Overall, the Cobas 6800 assay showed good concordance in detecting high-risk HPV in urine samples, which is consistent with HPV detection in clinician-obtained cervical samples [38].

The clinical performance of the assay was performed over a course of three days, and the results showed a high sensitivity, specificity, and accuracy of the urine-based high-risk HPV detection. The overall sensitivity was found to be 93% for HPV16, 94% for HPV18, and 90% for HPV68. The Cobas detected all three high-risk HPVs in spiked urine samples with 100% specificity with no false positive sample detected. Both the overall percent agreement and accuracy were calculated to be 95% for HPV16 and for HPV18, and 93% for HPV68. The sensitivity and specificity of this assay was comparable with HPV detection using clinician-obtained cervical samples [13,29,38]. Moreover, the results were also consistent with the sensitivity and specificity thresholds that are considered adequate for cervical screening [39]. To evaluate the diagnostic performance of HPV detection in urine samples of patients, a pilot study of 137 patients was performed. The assay effectively detected HPV in the first-void urine samples of 2 out of 110 males and 7 out of 63 females. The results of this pilot study, which was conducted using real clinical samples, indicate that the assay is compatible with the detection of HPV in urine samples and has the potential to be used for mass screening.

The stability and reproducibility of urine samples was also assessed in this study over a course of 7 days. The results showed a shift of approximately four Ct values observed over 7 days for HPV16, and β globulin, whereas a shift of only two Ct values was observed for HPV18 and HPV68, indicating a high stability of the urine samples. The reproducibility of the assay was assessed by calculating the coefficient of variation (CV%), which was found to be between 4.2 and 5.1% for HPV16, 1.9 and 3.6% for HPV18, 2.3 and 3.6% for HPV68, and 1.2 and 4.4% for β globulin, which lies within the acceptable range. The Cobas 6800 system also demonstrated high intra-laboratory reproducibility with urine samples when the assay was performed by two different technicians, which proves that the assay is compatible with first-void urine samples for detecting high-risk HPVs. The strength of this study was that the participants included both males and females. A significant amount of research shows that both cis and trans men are the main carriers for HPV and transfer the infection to women and other men through sexual contact. Therefore, HPV screening is equally important for men and non-binary people as for cisgender women [40,41].

## 5. Conclusions

In conclusion, the current study indicates the clinical utility and compatibility of self-collected first-void urine samples for high-risk HPV detection using the Cobas 6800 system. The assay showed high concordance, sensitivity, specificity, and reproducibility, which demonstrates that urine-based high-risk HPV detection using the Cobas 6800 system meets the requirements for usage in population-based cervical screening. Furthermore, this private, gender-inclusive, and non-invasive assay can circumvent the limitations of the Cobas 6800, and has the potential to increase the participation of young women, non-binary people, trans people, and cisgender men in screening programs. Urine-based HPV detection is cost-effective and well accepted, and can be used for mass screening of populations, allowing for the monitoring of vaccine effectiveness. This, in turn, can not only increase the screening coverage, but will also help the public understand more about HPV and the presence it has in our day-to-day lives.

## Figures and Tables

**Figure 1 vaccines-11-01071-f001:**
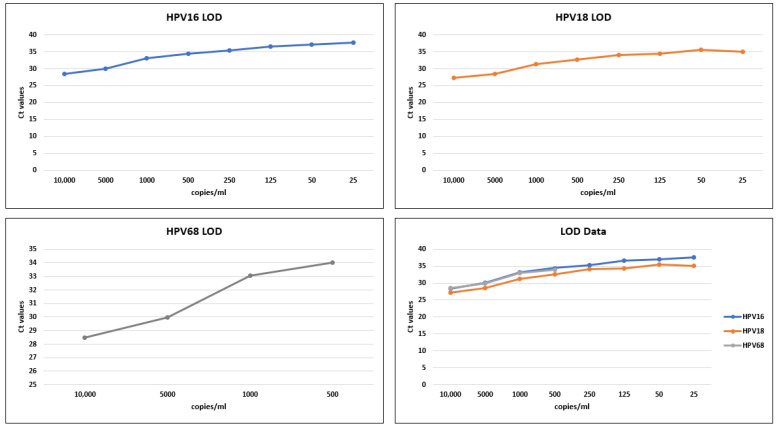
LOD for HPV16, HPV18, and HPV68.

**Table 1 vaccines-11-01071-t001:** Summary of clinical performance of Cobas 6800.

Total Samples/Calls	HPV 16	HPV 18	HPV 68
	120	120	90
True Positive	84	85	54
False Positive	0	0	0
True Negative	30	30	30
False Negative	6	5	6
Estimated Sensitivity	=100% × TP/(TP + FN)	=100% × TP/(TP + FN)	=100% × TP/(TP + FN)
	93%	94%	90%
Estimated Specificity	100% × TN/(FP + TN)	100% × TN/(FP + TN)	100% × TN/(FP + TN)
	100%	100%	100%
Overall Percent Agreement	=100% × (TP + TN)/Total	=100% × (TP + TN)/Total	=100% × (TP + TN)/Total
	95%	95%	93%
Accuracy	95%	95%	93%

TP: True Positive, TN: True Negative, FP: False Positive, FN: False Negative.

**Table 2 vaccines-11-01071-t002:** Pilot validation study using clinical urine samples.

	Clinical Samples	Positive	Negative	Invalid
Male	110	2	108	0
Female	63	7	56	0
Total Samples	173	9	164	0

**Table 3 vaccines-11-01071-t003:** Detailed table of stability and reproducibility analysis on 20 samples over a period of 7 days. Neg: Negative, IM: Instrument Malfunction.

**HPV16**								**HPV18**						
**Samples**	**Day 1**	**Day 2**	**Day 3**	**Day 4**	**Day 5**	**Day 6**	**Day 7**	**Day 1**	**Day 2**	**Day 3**	**Day 4**	**Day 5**	**Day 6**	**Day 7**
Sample 1	30.09	31.49	32.05	32.72	IM	34.58	34.09	28.49	28.9	29.35	29.78	IM	30.99	31.16
Sample 2	30.39	31.45	31.74	32.56	IM	33.99	33.56	28.86	29.21	29.43	29.64	IM	29.93	30.48
Sample 3	30.24	31.41	31.92	32.74	IM	34.82	33.64	28.68	28.77	29.33	29.76	IM	30.66	30.89
Sample 4	30.56	31.87	31.74	32.67	IM	34.7	33.63	28.72	29.07	29.31	29.97	IM	30.41	30.59
Sample 5	30.46	31.49	32.09	32.74	IM	34.79	33.73	28.77	29.3	29.4	29.77	IM	30.95	30.54
Sample 6	30.28	31.43	31.88	32.79	33.44	34.37	33.73	28.74	28.89	29.34	29.98	30.61	30.78	30.8
Sample 7	30.33	31.1	31.84	32.71	33.26	35.13	33.79	28.66	28.46	29.35	29.94	30.15	30.71	30.84
Sample 8	30.12	31.42	31.84	32.59	33.47	34.3	33.73	28.8	29.1	29.14	30.12	30.94	30.64	30.63
Sample 9	30.29	31.28	31.69	32.47	33.05	34.84	34.1	28.87	28.7	29.17	29.57	30.23	30.9	31.12
Sample 10	30.79	31.37	31.61	32.53	32.96	34.83	33.42	29.14	29.18	29.1	29.86	30.21	31.01	30.6
Sample 11	30.32	31.44	31.83	32.69	33.43	34.99	33.73	28.49	28.9	29.05	29.69	29.97	30.49	30.85
Sample 12	30.38	31.74	31.68	32.47	33.51	35.08	33.54	28.72	29.34	29.27	29.49	30.36	30.88	31.18
Sample 13	30.28	31.17	31.69	32.55	33.43	34.99	33.71	28.76	28.81	29.19	29.51	30.97	30.88	30.83
Sample 14	30.31	31.4	31.64	32.36	33.35	34.84	33.91	28.66	28.9	29.18	29.68	30.47	30.3	30.69
Sample 15	30.16	31.53	31.69	32.45	33.25	34.9	33.63	28.51	29.1	29.19	29.62	30.09	30.87	30.58
Sample 16	Neg	Neg	Neg	Neg	Neg	Neg	Neg	Neg	Neg	Neg	Neg	Neg	Neg	Neg
Sample 17	Neg	Neg	Neg	Neg	Neg	Neg	Neg	Neg	Neg	Neg	Neg	Neg	Neg	Neg
Sample 18	Neg	Neg	Neg	Neg	Neg	Neg	Neg	Neg	Neg	Neg	Neg	Neg	Neg	Neg
Sample 19	Neg	Neg	Neg	Neg	Neg	Neg	Neg	Neg	Neg	Neg	Neg	Neg	Neg	Neg
Sample 20	Neg	Neg	Neg	Neg	Neg	Neg	Neg	Neg	Neg	Neg	Neg	Neg	Neg	Neg
Average	30.33	31.43	31.8	32.6	33.32	34.74	33.73	28.72	28.98	29.25	29.76	30.4	30.69	30.78
**HPV68**								**β globulin**						
Samples	**Day 1**	**Day 2**	**Day 3**	**Day 4**	**Day 5**	**Day 6**	**Day 7**	**Day 1**	**Day 2**	**Day 3**	**Day 4**	**Day 5**	**Day 6**	**Day 7**
Sample 1	29.96	30.44	30.99	31.47	IM	32.63	32.72	30	31.21	31.13	31.58	N/A	N/A	32
Sample 2	30.08	30.76	30.98	31.42	IM	31.56	32.22	30.09	31.29	30.77	31.56	N/A	32.7	31.41
Sample 3	29.9	30.24	31.02	31.58	IM	32.65	32.45	29.73	30.98	30.72	31.33	N/A	33.8	32.07
Sample 4	30.29	30.67	30.81	31.68	IM	32.51	32.22	30.05	31.54	30.78	31.46	N/A	33.26	31.85
Sample 5	30.29	30.7	31.15	31.46	IM	32.76	32.32	30.69	31.15	30.78	31.1	N/A	33.94	31.66
Sample 6	30.21	30.46	31.01	31.64	32.28	32.31	32.3	30.08	31.27	31.05	31.63	31.97	33.61	31.87
Sample 7	30.05	30.14	31.13	31.65	32.05	32.74	32.19	29.68	30.61	31.32	31.25	31.59	33.82	32.27
Sample 8	29.88	30.68	30.81	31.63	32.36	32.55	32.4	29.55	30.79	30.51	31.18	32.03	33.42	32.35
Sample 9	30.07	30.28	30.99	31.32	32.07	32.72	32.7	29.88	30.32	30.62	31.24	31.77	N/A	32.73
Sample 10	30.44	30.35	30.63	31.39	31.89	32.72	32.46	30.46	29.18	30.58	31.41	31.49	N/A	31.66
Sample 11	30.03	30.52	31.1	31.44	31.85	32.39	32.34	30.12	31	30.73	31.81	31.86	33.7	32.54
Sample 12	30.22	30.8	30.88	31.41	32.27	32.65	32.22	30.29	31.84	30.39	31.75	32.47	33.77	31.75
Sample 13	30.14	30.64	30.87	31.47	32.04	32.64	32.3	29.92	31.39	30.56	31.35	32.34	33.7	31.86
Sample 14	29.98	30.31	30.57	31.26	32.3	32.57	32.52	29.64	31.13	30.55	31.6	32.47	33.59	32.09
Sample 15	29.88	30.58	30.97	31.39	31.99	32.7	31.66	29.36	30.83	31.11	31.56	31.48	N/A	32.16
Sample 16	Neg	Neg	Neg	Neg	Neg	Neg	Neg	30.79	31.84	32.02	31.51	32.1	31.27	31.5
Sample 17	Neg	Neg	Neg	Neg	Neg	Neg	Neg	31.65	31.52	31.65	31.09	32.7	31	32.18
Sample 18	Neg	Neg	Neg	Neg	Neg	Neg	Neg	31.77	31.2	31.23	31.51	32.3	31.82	32
Sample 19	Neg	Neg	Neg	Neg	Neg	Neg	Neg	31.28	31.09	32.16	29.7	32.12	31.72	31.72
Sample 20	Neg	Neg	Neg	Neg	Neg	Neg	Neg	30.93	31.82	31.23	32.46	32.03	31.72	31.3
Average	30.09	30.5	30.92	31.48	32.11	32.54	32.33	29.96	30.97	30.77	31.4	31.9	33.57	32

**Table 4 vaccines-11-01071-t004:** Comparison of ThinPrep and urine-based HPV detection using Cobas 6800.

	HPV16		HPV18		HPV68	
Sample Type	ThinPrep	Urine	ThinPrep	Urine	ThinPrep	Urine
Total Samples/Calls	30	120	30	120	30	90
True Positive	30	84	30	85	30	54
False Positive	0	0	0	0	0	0
True Negative	30	30	30	30	30	30
False Negative	0	6	0	5	0	6
Sensitivity = 100% × TP/(TP + FN)	100%	93%	100%	94%	100%	90%
Specificity = 100% × TN/(FP + TN)	100%	100%	100%	100%	100%	100%
Accuracy = 100% × (TP + TN)/Total	100%	95%	100%	95%	100%	93%
Overall Percent Agreement	100%	95%	100%	95%	100%	93%

## Data Availability

Not applicable.

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
