# Peer review of "Clinical Performance of Cobas 6800 for the Detection of High-Risk Human Papillomavirus in Urine Samples"

_vaccines, 2023, doi:10.3390/vaccines11061071_

Round 1
Reviewer 1 Report
The authors propose that universal HPV screening be done via PCR using the COBAS 600 system and urine samples. To evaluate this methodology, the authors spike urine samples with HPV variants and then submit the samples for analysis using the COBAS 600 system. While the idea of identifying a universal method of HPV screening is laudable, the approach the authors use to provide proof of concept is not poor.
1-What were the patient parameters for those contributing urine to the study?
2-For the spiked and negative-control samples, the authors used urine from patients. Given the authors are looking for HPV, what was done to ensure HPV was not already present prior to spiking?
3-Given the sensitivity of PCR, I am not surprised the spiked urine was positive. The real question is whether patients with HPV--whether it be oral, cervical, etc.--would be positive via a urine test--i.e. is HPV from these various sites shed in urine?? Given the authors were trying to demonstrate that HPV could be detected in urine, why did the authors not obtain urine samples from known HPV-positive patients and see whether the urine was positive for HPV? i.e. how do the authors even know that HPV will be present in urine of HPV positive individuals??
4-Line 129, define LOD. Any time an acronym is first used, it must be defined and then thereafter can be used as an abbreviation.
5-Lines 151-156, the authors indicate they evaluated urine from patients with urinary and reproductive issues. The data is presented in Table 3. This study needs to be included in the Methods section. Further, why did the authors do this study and not examine urine from patients with known HPV??
6-Confirm that Methods are written in such a way as to account for all data presented and that if one wanted to reproduce the study presented, that they could do so, using the methodology included within the manuscript.
7-Lines 191-192, the authors state, "It has also to be noted that the urinary tract, vagina, cervix and vulva, lies in proximity with each other." Just because tissues are proximal to the urinary tract does not mean that HPV will be shed in urine. Further, this sort of negates the authors goal to find a method of testing that takes into all locations in which HPV might be found, including more distant sites such as the oral cavity.
8-Finally, authors need to have someone carefully check punctuation, grammar, sentence construction and formatting.
Author Response
The authors propose that universal HPV screening be done via PCR using the COBAS 600 system and urine samples. To evaluate this methodology, the authors spike urine samples with HPV variants and then submit the samples for analysis using the COBAS 600 system. While the idea of identifying a universal method of HPV screening is laudable, the approach the authors use to provide proof of concept is not poor.
1-What were the patient parameters for those contributing urine to the study?
Reply: Patient parameters:
- First void urine.
- For spiked samples the urine was first tested to make sure the sample was HPV negative. Only HPV negative samples were used.
2-For the spiked and negative-control samples, the authors used urine from patients. Given the authors are looking for HPV, what was done to ensure HPV was not already present prior to spiking?
Reply: To address your concern regarding the urine samples, we first tested all urine samples for HPV using the COBAS 6800 system prior to spiking with known HPV variants or using them as negative controls. Only those samples that tested negative for HPV were used in the study.
3-Given the sensitivity of PCR, I am not surprised the spiked urine was positive. The real question is whether patients with HPV--whether it be oral, cervical, etc.--would be positive via a urine test--i.e. is HPV from these various sites shed in urine?? Given the authors were trying to demonstrate that HPV could be detected in urine, why did the authors not obtain urine samples from known HPV-positive patients and see whether the urine was positive for HPV? i.e. how do the authors even know that HPV will be present in urine of HPV positive individuals??
Reply: To address this, we conducted a clinical study where we obtained urine samples from known HPV-positive patients and tested them using the COBAS 6800 system. As you correctly noted, HPV can be found in various sites, including the oral and cervical cavity, and we wanted to see if HPV could be detected in urine samples from naturally infected individuals. Our clinical study included 173 patients, and we were able to detect HPV in nine of these patients. These results are shown in Table 3 of the manuscript. Our findings suggest that HPV can indeed be detected in urine samples from HPV-positive individuals, further supporting the use of urine samples for HPV screening.
4-Line 129, define LOD. Any time an acronym is first used, it must be defined and then thereafter can be used as an abbreviation.
Reply: Authors agree with the observation and made appropriate changes by defining the LOD in the abstract section (line 24).
5-Lines 151-156, the authors indicate they evaluated urine from patients with urinary and reproductive issues. The data is presented in Table 3. This study needs to be included in the Methods section. Further, why did the authors do this study and not examine urine from patients with known HPV??
Reply: The authors agree with the observation and made appropriate changes by including the pilot study using patients samples in the Methods section. (Line 106-110). Moreover, since there was unavailability of known HPV patient samples that is why authors opted for blind clinical.
6-Confirm that Methods are written in such a way as to account for all data presented and that if one wanted to reproduce the study presented, that they could do so, using the methodology included within the manuscript.
Reply: We acknowledge that the complete details of the methodology are not currently available due to pending patent applications. However, we assure you that we are committed to sharing the complete details of the methodology once the patent is completed. We understand the importance of reproducibility in research and will make every effort to ensure that the methodology can be replicated by other researchers in the future.
7-Lines 191-192, the authors state, "It has also to be noted that the urinary tract, vagina, cervix and vulva, lies in proximity with each other." Just because tissues are proximal to the urinary tract does not mean that HPV will be shed in urine. Further, this sort of negates the authors goal to find a method of testing that takes into all locations in which HPV might be found, including more distant sites such as the oral cavity.
Reply: Thank you for bringing this to our attention. We agree with your point that the proximity of tissues to the urinary tract does not necessarily mean that HPV will be shed in urine. In response to your feedback, we have removed this part from the original manuscript to avoid any misunderstanding (Line 195-197).
8-Finally, authors need to have someone carefully check punctuation, grammar, sentence construction and formatting.
Reply: Manuscript is edited by a professional and native English editor from UK.
Reviewer 2 Report
An important piece of work in the steps moving forward.
Author Response
Thank You for your valuable comments
Reviewer 3 Report
The paper presented by Brian Joseph Hajjar and coworkers is potenzially very important. However, there are several points that need review before it can be considered for publication. The most critical point are material and methods. In particular, there is no reference to an official validation method. The described technique is not a validation but a preliminary study. The extraction of a spiked urine sample is not comparable to that of a naturally positive sample. Viral DNA cauld be inside the cell and not inside of liquid fraction of urine. The authors should test patients with the classical method and those who tested positive also analyze them with the method they describe and calculate accuracy sensitivity and specificity on these data. They tested samples from patients who had historical problems but did not demonstrate the applicability of the method so they would have to do the two tests at the same time. Moreover, it is also unclair how HPV samples were spiked to assess LOD. H2O2? unire? other?
Author Response
The paper presented by Brian Joseph Hajjar and coworkers is potenzially very important. However, there are several points that need review before it can be considered for publication. The most critical point are material and methods. In particular, there is no reference to an official validation method. The described technique is not a validation but a preliminary study. The extraction of a spiked urine sample is not comparable to that of a naturally positive sample. Viral DNA cauld be inside the cell and not inside of liquid fraction of urine. The authors should test patients with the classical method and those who tested positive also analyze them with the method they describe and calculate accuracy sensitivity and specificity on these data. They tested samples from patients who had historical problems but did not demonstrate the applicability of the method so they would have to do the two tests at the same time. Moreover, it is also unclair how HPV samples were spiked to assess LOD. H2O2? unire? other?
Reply: Thank you for your comments on our manuscript. We would like to respond to your concern regarding the validation of our method.
Firstly, we would like to clarify that the validation of our method was conducted in accordance with the guidelines of the New Jersey Department of Health. We received approval from the department to conduct HPV testing via urine using the COBAS 6800 system. Secondly, the thinprep is an FDA-approved test for HPV using cervical swabs. However, we modified the test to enable HPV detection in urine, which overcomes the invasiveness of the sampling procedure. Moreover, to the best of our knowledge, our study is the first of its kind to perform COBAS 6800-based HPV testing using urine samples. We understand the importance of validation in ensuring the accuracy and reliability of our method, and we believe that our study provides a solid foundation for further validation in larger clinical trials.
We understand your concern about the comparability of spiked urine samples and naturally positive samples. We would like to assure you that we conducted a clinical study to further evaluate the presence of HPV in urine samples. In our clinical study, we blindly collected urine samples from 173 patients. The samples were analyzed using the COBAS 6800 system, and we were able to detect HPV in 9 out of 173 positive samples. This confirms the presence of HPV in urine samples from naturally positive patients.
Regarding your concern about the cellular and cell-free DNA detection, we would like to clarify that the COBAS 6800 system used in our study is designed to detect both cell-associated and cell-free DNA. Therefore, it is capable of detecting HPV DNA from both cellular and non-cellular fractions of urine samples.
It must be noted that finding HPV positive patients is challenging. Therefore, we conducted a study using spiked samples with the FDA approved method, and we achieved approximately 90% concordance between the spiked samples using the COBAS 6800 system. Nonetheless, we will continue to explore opportunities to further validate our proposed method in the future.
Regarding your concern about the method of spiking urine samples and making dilutions. The positive controls for HPV variants were obtained from Exact Diagnostics and they provide the controls in a liquid medium (PreservCyt®). The controls were then subjected to a series of dilutions using negative urine volumes to obtain different concentrations of the positive control. The dilution factor was calculated based on the volume of the positive control and the volume of negative urine used for the dilution.
Round 2
Reviewer 3 Report
Thi is not a validation study so this statement must be removed in all test. This is a first report on the possibility to use Cobas on urine samples.
My major note is on the clinical samples. Authors claim to have tested 174 samples and 9 positive to HPV. The same samples should be test to HPV with classical methods e.g. RT-PCR and Cobas. Than results should be showed e compared. With out this results this study is not robust
Round 3
Reviewer 3 Report
The authors have improved the study and have answer to my questions.